# Correlation Analysis of Soil Microbial Communities and Physicochemical Properties with Growth Characteristics of *Sageretia thea* Across Different Habitats

**DOI:** 10.3390/plants13233310

**Published:** 2024-11-26

**Authors:** Dae-Hui Jeong, Yeong-Bae Yun, Ho-Jun Son, Yurry Um, Jeong-Ho Song, Jiah Kim

**Affiliations:** Forest Medicinal Resources Research Center, National Institute of Forest Science, Yeongju-si 36040, Republic of Korea; najdhda@korea.kr (D.-H.J.); yybangel@korea.kr (Y.-B.Y.); urspower@korea.kr (Y.U.); sjh8312@korea.kr (J.-H.S.); jiahkim@korea.kr (J.K.)

**Keywords:** bacteria, correlation analysis, growth characteristics, *Sageretia thea* (Osbeck) M.C.Jonst., soil chemical

## Abstract

This study aimed to investigate the growth characteristics of *Sageretia thea* and analyze the correlations between soil physicochemical properties and microbial communities in its native habitats. Soil physicochemical properties were characterized by organic matter (0.37–36.43%), available phosphate (57.96–315.90 mg/kg), potassium (0.11–1.17 cmol^+^kg^−1^), calcium (1.23–25.97 cmol^+^kg^−1^), magnesium (0.43–15.01 cmol^+^kg^−1^), sodium (0.04–6.16 cmol^+^kg^−1^), and pH (4.68–7.05), indicating slightly acidic to neutral conditions. *S. thea* exhibited variable growth characteristics across habitats; leaf length and width were largest in Jangnam-ri and Hacka-ri, respectively, whereas Docheong-ri promoted higher fruit growth. The soil microbial community composition was dominated by *Proteobacteria*, *Actinobacteria*, and *Acidobacteria* at the phylum level (76.09%) and by *Alphaproteobacteria*, *Actinobacteria_c*, and *Vicinamibacter_c* at the class level (40%). Soil physicochemical properties were significantly correlated with Actinobacteria, *Acidobacteria*, and *Chloroflexi* at the phylum level, and all microbial groups except *Spartobacteria* at the class level. *Furthermore*, growth characteristics were significantly correlated with all microbial communities except *Acidobacteria* and *Firmicutes* at the phylum level, and *Acidobacteria*, *Thermoleophilia*, and *Rubrobacteria* at the class level. These findings provide a foundation for developing efficient cultivation techniques for *S. thea* based on its soil microbiome and habitat conditions.

## 1. Introduction

Bacteria represent the most abundant microorganisms in soil, with over 100 million individuals across 10^4^–10^6^ species per gram of soil, and the total number of microorganisms, including bacteria, is estimated to exceed 10^8^ species per gram of soil [1]. Microorganisms play a pivotal ecological role in regulating biogeochemical cycles and influence plant growth [2]. Their abundance and diversity are regulated by various physical and chemical properties of the soil, such as pH, organic matter, P and K content, and meteorological conditions, such as temperature, precipitation, and light [3,4]. Microbial communities are sensitive to environmental changes and serve as indicators of soil conditions [5]. Therefore, studies have been conducted to screen microbial communities and analyze the soil microbiome of native or cultivated lands for optimizing growing conditions for crops, identifying the causes of crop damage in monocropping systems [6,7,8,9].

(Osback) M. C. Johnst., a perennial shrub belonging to the family Rhamnaceae, is represented by a single genus and species in Korea [10]. This species is distributed along the southern coast and Jeju island. The extracts derived from this plant have diverse medicinal potential; its branch and leaf extracts reduce the viability of human colorectal cancer cells, inducing cyclin D1 proteasomal degradation and HO-1 expression [11], silver nanoparticles synthesized from its root extract exhibit antioxidant effects [12], and flavonoids and phenolic acids extracted from its above-ground parts contribute to the improvement and prevention of degenerative diseases [13]. Additionally, fruit extracts of this plant have antioxidant and immune response-modulation effects [14,15], while callus extracts are used in cosmetics to inhibit reactive oxygen species and IL-8 production induced by fine dust [16]. Related species such as *S. gracilis* are popular as ornamental plants for bonsai gardening in China [17]. Notably, despite its substantial potential, *S. thea* is primarily collected from natural environments with a lack of established cultivation techniques. Therefore, there is a need to cultivate *S. thea*, a wild plant, because its anti-cancer, anti-inflammatory, and immune response-modulating properties are expected to be useful and profitable, and there is a lack of research on its cultivation.

Most plants on Earth are associated with soil microorganisms throughout their life cycle [18]. For example, the genera *Pseudomonas* and *Bacillus* dissolve insoluble phosphate compounds into an absorbable form, supplying plants with essential nitrogen and phosphorus for their growth [19]. Microorganisms such as *Glomus intraradices*, *G. claroideum*, *Gigaspora margarita*, and others enhance plant growth by improving water and nutrient uptake [20].

In addition, *Bacillus* species are antagonistic to plant pathogens, such as *Botrytis cinerea*, *Rhizoctonia solani*, and *Sclerotinia minor*, and have been shown to promote plant growth [21]. In particular, orchids are closely related to soil microbes; for non-photosynthetic orchids, interactions with orchid mycorrhizal fungi are crucial for seed germination, early chlorophyll production, and meeting their nutritional requirements [22,23]. Collectively, various soil microorganisms exert profound effects on plant growth, and analyzing their interactions with plants is imperative for the stable production and efficient management of plants.

In this study, we aimed to analyze the native soil microbiome of *S. thea*, which has substantial medicinal, cosmetic, and ornamental potential. We examined correlations between the soil microbiome, soil physicochemical properties, and growth characteristics to establish data that will inform the development of efficient cultivation models.

## 2. Results

### 2.1. Soil Physicochemical Properties

The soil physicochemical properties of *S. thea* habitats were analyzed and found to be slightly acidic to neutral, with a pH of 4.68–7.05 in sandy loam (BS, CD, MS), loamy loam (DC, ND), and loamy sand (DH, HG, JN, JJ, HY). The organic matter (OM) ranged from 0.37% in BS to 36.43% in HY, total nitrogen ranged from 0.03% in BS to 1.39% in HY, available phosphate (Avai. P) ranged from 57.96 mg/kg in the BS to 315.90 mg/kg in the ND group. In addition, potassium (K) was observed as 0.11–1.17 cmol^+^kg^−1^, calcium (Ca) as 1.23–25.97 cmol^+^kg^−1^, magnesium (Mg) as 0.43–15.01 cmol^+^kg^−1^, and sodium (Na) as 0.04–6.16 cmol^+^kg^−1^. Furthermore, cation exchange capacity (CEC) was observed as 4.15–50.88 cmol^+^kg^−1^, and electrical conductivity (EC) as 0.11–1.78 dS m^−1^ (Table 1).

### 2.2. Growth Characteristics

Table 2 presents the growth characteristics of *S. thea*. *S. thea* showed regional differences in growth characteristics; leaf length was 38.15 mm in JN and width was 22.55 mm in HG. The fruit growth characteristics were the highest in DC, with a 7.87 mm length, 8.95 mm width, and 0.41 g fresh weight compared to other regions. The sweetness was 19.81 °Brix in MS, and the hardness was 1.56 N in ND, which was the highest value measured.

### 2.3. Soil Microbial Community

The analysis of the relative abundance of microbial communities in the soil of *S. thea* habitats demonstrated that *Proteobacteria* dominated at the phylum level and *Alphaproteobacteria* dominated at the class level in all habitats (Figure 1). In terms of phylum, *Proteobacteria* was identified as the most dominant phylum, with an average 34.83%, with the highest value for DH (45.83%) and the lowest for HY (29.74%). *Actinobacteria* had an average of 21.31% and *Acidobacteria* had an average of 19.95%; three microbial phyla accounted for three-quarters of the total (76.09%) of all microbial phyla. And *Verrucomicrobia* accounted for 6.41%, followed by *Firmicutes* (4.48%), *Chloroflexi* (4.25%), and *Bacterioidetes* (3.91%), which represented 3–4% of the total; *Gemmatimonadetes* (2.22%), *Planctomycetes* (2.08%), *Nitrospirae* (1.48%), *Cyanobacteria* (1.40%), and *Laterscibacteria*_WS3 (1.11%) showed an average of 1–2% (Figure 1A).

At the class level, *Alphaproteobacteria* had the highest value, with an average of 21.02%, followed by *Actinobacteria_c*, with an average of 10.54%, and *Vicinamibacter_c*, with an average of 7.31%, resulting in three microbial classes accounting for 38.87%. This was followed by *Spartobacteria* (5.22%). *Acidobacteria* (5.06%), and *Betaproteobacteria* (5.02%), which accounted for 5%, and *Solibacteres* (4.65%), *Gammaproteobacteria* (4.47%), and *Rubrobacteria* (4.26%) accounted for 4%; *Deltaproteobacteria* (3.96%), *Themoleophilia* (3.79%), *Blastocatellia* (3.43%) accounted for 3%, and others accounted for 1–2% (Figure 1B).

### 2.4. Correlation Between Soil Microbial Community and Soil Properties

Duncan’s correlation analysis between soil microbial communities and soil properties was performed to determine the effect of soil properties on soil microbial communities (Appendix A and Figure 2). At the phylum level, microbial communities showed significant correlations, except for *Chloroflexi* and *Firmicutes*, which showed different behaviors between *Acidobacteria* and *Actinobacteria*. *Acidobacteria* exhibited a negative correlation with all soil properties, showing statistical significance for all except P (r = −0.247, *p* = 0.094), Na (r = −0.251, *p* = 0.090), and CEC (r = −0.037, *p* = 0.423). In contrast, *Actinobacteria* demonstrated a positive correlation with most soil heterogeneities, being significant for all except P (r = 0.265, *p* = 0.078) and CEC (r = 0.041, *p* = 0.415).

When correlations were analyzed at the class level, *Alphaproteobacteria* demonstrated a significant negative correlation with CEC (r = −0.622, *p* = 0.000), while *Actinobacteria_c* was significantly negatively correlated with Ca (r = 0. 398, *p* = 0.029), pH (r = 0.378, *p* = 0.040), and EC (r = 0.430, *p* = 0.018), and *Vicinamibacter_c* showed a significant and positive correlation with CEC (r = 0.631, *p* = 0.000). *Acidobacteria* were negatively correlated with Ca (r = −0.574, *p* = 0.008) and pH (r = −0.619, *p* = 0.004), *Betaproteobacteria* were negatively correlated with OM (r = −0.386, *p* = 0.035) and pH (r = −0.408, *p* = 0.025), and *Solibacteres* were negatively correlated with K (r = −0. 396, *p* = 0.033), Ca (r = −0.626, *p* = 0.001), CEC (r = −0.643, *p* = 0.000), and EC (r = −0.400, *p* = 0.031). In contrast, *Gammaproteobacteria* and *Thermoleophilia* showed a strong positive correlation with organic matter, total nitrogen, and pH, including trace elements such as calcium and magnesium, which is in contrast to *Acidobacteria*, *Betaproteobacteria*, and *Solibacteres*.

### 2.5. Correlation Between Soil Microbial Community and Growth Characteristics

Duncan’s correlation analysis between the growth characteristics of *S. thea* and microbial communities in the soil revealed that at the phylum level, all communities, except *Acidobacteria* and *Firmicutes,* were significantly correlated (Appendix A and Figure 3). *Proteobacteria* was positively correlated with leaf width (*r* = 0.344, *p* = 0.031) and fruit weight (*r* = 0.321, *p* = 0.042), *Verrucomicrobia* with fruit width (*r* = 0.382, *p* = 0.019), and *Actinobacteria* with sweetness (*r* = 0.361, *p* = 0.025); *Bacteroidetes* was negatively correlated with leaf length (*r* = −0.373, *p* = 0.025) and width (*r* = −0.319, *p* = 0.049), and *Chloroflexi* demonstrated a significant negative correlation with fruit growth characteristics [length (*r* = −0.359, *p* = 0.028), width (*r* = 0.470, *p* = 0.005), and weight (*r* = 0.476, *p* = 0.004)] and a significant positive correlation with hardness (*r* = 0.330, *p* = 0.040). At the class level, we found a positive correlation between *Acidobacteria* and leaf length (*r* = 0.447, *p* = 0.048), *Thermoleophilia* and sweetness (*r* = 0.380, *p* = 0.042), and *Rubrobacteria* was negatively correlated with fruit weight (*r* = −0.377, *p* = 0.044).

## 3. Discussion

### 3.1. Soil Physicochemical Properties of S. thea Habitats

*S. thea* is a plant resource native to Korea, China, Japan, India, Taiwan, and Vietnam, and is expected to be utilized for medicinal and ornamental purposes [24]. Therefore, this study was conducted to obtain basic data for developing efficient cultivation technology. A previous study on the soils of *S. thea* habitats found that sandy loam soils, characterized by a high percentage of sand, were the most prevalent in these areas [25]. This study also confirmed similar results to the previous study’s report of a high proportion of sand and a low proportion of silt and clay, which differed from the average forest soil in Korea (41.7% sand, 41.5% silt, and 16.8% clay) [26]. In addition, it was found that there were differences in trace elements such as Ca and Mg, including OM, among the sites, especially of Na, which was 6.16 cmol^+^kg^−1^ in HY compared to 0.04–0.51 cmol^+^kg^−1^ in most sites. HY is located on the coast of Jeju Island, South Korea, and is a representative island region with a subtropical oceanic climate [27]. In other words, compared with other sites, it is believed that the high Na values are due to geographical characteristics that are directly affected by strong winds and waves caused by the oceanic climate.

### 3.2. Growth Characteristics of S. thea Across Different Habitats

In this study, we identified differences in *S. thea* growth among sites. Plant growth is not driven by a single factor but by complex interactions between many environmental factors, including weather conditions, soil chemistry, and microbial communities [28,29,30]. This study was conducted at various native sites of *S. thea* from Jeolla Province to Jeju Island, Korea. In other words, there are differences in latitude (HY: 33°14′–BS: 34°52′) and longitude (BS: 126°01′–ND: 128°32′) of each study site, and consequently differences in meteorological characteristics such as average annual temperature (2.2 °C) and precipitation (400 mm) due to geographical characteristics [31]. In addition, the interaction of various environmental factors, such as the soil characteristics at each site identified earlier, may have contributed to the differences in *S. thea* growth examined in this study.

### 3.3. Abundance of Soil Microbial Communities in S. thea Habitats

Soil microorganisms have a general influence on the life of plants, particularly the uptake of inorganic elements (Mg, Na, Ca, etc.) required for growth from the soil [30]. Therefore, soil microbial community analysis was conducted for each study site, and it was found that *Proteobacteria* dominated at the phylum level, *Alphaproteobacteria* dominated at the class level, and the sum of the abundances of *Proteobacteria*, *Acidobacteria*, and *Actinobacteria* phyla accounted for 76.09% at the phylum level. *Proteobacteria*, *Acidobacteria*, and *Actinobacteria* are the dominant phyla of microorganisms in most soils and are highly abundant in coastal areas of Korea and most coastal areas of East Asian countries, such as China and Japan [32,33,34,35], and *Alphaproteobacteria* are also the most abundant class of microorganisms in the distribution of marine microorganisms [6,36]. It is known that the above microbial communities show a close correlation with the metallic composition of soils (Fe, Cu, As, etc.) and are highly abundant in most soils, especially in low-pH and polluted environments [37,38,39,40]. These characteristics imply a high adaptation and resilience of the microbial community to the surrounding environment and may also explain the native environment of *S*, *thea*, which is distributed in coastal areas in slightly acidic to neutral soils with a low pH.

### 3.4. Effect of Soil Physicochemical Properties on Microbial Communities

The soil environment affects the distribution and abundance of microorganisms, as has been recognized in various studies [41]. Therefore, in this study, we analyzed the correlation between soil physicochemical properties and microbial communities, and identified different patterns of *Acidobacteria* and *Actinobacteria* at the phylum level. *Acidobacteria* were negatively correlated with most soil chemistries, whereas *Actinobacteria* were positively correlated. Both *Acidobacteria* and *Actinobacteria* are major microbial communities present in the soil, and each has its own characteristics and roles in the ecosystem. *Acidobacteria* can live over a wide range of soil pH but are particularly adaptable to acidic environments [42]. They are primarily responsible for cycling organic matter in the soil and enhancing the availability of various nutrients [43]. Studies involving *Acidobacteria* have shown that their abundance is strongly and negatively correlated with soil organic carbon [44]. These are not all subgroups within this phylum, but many microbes are oligotrophic groups that thrive in low-nutrient environments [43]. *Actinobacteria*, on the other hand, are predominantly found in neutral or slightly alkaline soils [45] and perform ecological functions, such as antibiotic action and pathogen inhibition [46]. They also show a strong positive correlation with soil moisture and temperature, including available nitrogen and organic carbon [47]. The two microbial communities exhibit differences in their ecological roles and environmental adaptations. However, both taxa play important roles in the soil, such as organic matter cycling and antibiotic activity, and their distribution in the soil is strongly correlated with soil chemistry, such as with pH, organic matter content, and nutrient concentration. This suggests that the distribution and activity of both communities are regulated by the soil properties and play important roles in soil ecosystems. At the class level, CEC was strongly correlated with diverse microbial communities (*Alphaproteobacteria*, *Vicinamibacteria_c*, *Acidobacteriia*, *Solibacteres*, *Deltaproteobacteria*, and *Blastocatellia*). CEC is an indicator of the ability of soil to absorb and exchange cations (Ca, Mg, K, etc.), and soils with a higher CEC have an increased ability to absorb and retain nutrients [48,49]. In other words, CEC affects the distribution and abundance of soil microbial communities, which has been shown in various previous studies [50,51,52], and our results confirmed this through a correlation analysis between soil physicochemical properties and microbial communities.

### 3.5. Effect of Microbial Communities on Growth Characteristics of S. thea

The soil microbial community is influenced by a variety of environmental factors and interacts closely with plant growth. In this study, we characterized the growth and interactions of *S. thea* within the microbial community phyla *Proteobacteria* and *Chloroflexi*. *Proteobacteria* abundance was significantly and positively correlated with leaf width and fruit weight. *Proteobacteria* are found in soils under various conditions and are involved in the decomposition of soil OM and the biogeochemical cycling of carbon, nitrogen, and sulfur [53]. This microbial community is called plant growth-promoting rhizobacteria and is closely related to plant growth [54]. The soil microbiota plays a crucial role in plant growth through interactions that influence nutrient absorption, disease resistance, and environmental stress tolerance. Soil microorganisms such as plant growth-promoting rhizobacteria (PGPR) and mycorrhizal fungi can alter soil structure and the availability of essential nutrients like nitrogen (N) and phosphorus (P), promote root development, and enhance resistance to pathogens, thereby facilitating nutrient uptake [43]. In addition, studies on Arabidopsis have confirmed its ability to increase plant growth by enhancing the availability of phosphorus (P), which is involved in cell membrane formation and energy-dependent metabolic processes, and that of iron (Fe), which is involved in cellular redox functions such as photosynthesis and respiration [55,56,57]. In other words, the microorganisms in *Proteobacteria* transform nutrients in the soil needed for plant growth into a form that can be absorbed by plant roots, which in turn affects plant growth. This is why they are positively correlated with leaf and fruit growth characteristics.

In general, *Chloroflexi* can improve soil fertility through ecological roles, such as the decomposition of organic matter in the soil and nitrogen fixation, and can provide the nutrients (nitrogen, phosphorus, etc.) necessary for fruit growth [33,58]. However, in this study, *Chloroflexi* was negatively correlated with overall fruit growth characteristics (length, width, and weight). Although studies have been conducted on the effects of different soil microbial communities on plant fruit growth, a clear correlation between *Chloroflexi* and fruit growth is not known [59,60,61]. In addition, plant growth can be affected by many variables, such as environment, soil type, and plant species [62]. Therefore, it is necessary to study the effects of *Chloroflexi* on plant fruit growth. Once the specific mechanisms are identified, we will have a clearer understanding of how these microbial communities affect fruit growth.

## 4. Materials and Methods

### 4.1. Study Habitat Site Selection and Collecting Soil Samples

A total of 10 *S. thea* sites were selected in Jeolla Province and Jeju Island, Republic of Korea (Table 3, Figure 4), and 100 g of rhizosphere soil samples from 10 to 30 cm of soil core was collected in May 2023 after removing the soil topsoil to collect soil samples from the study sites. The soil samples were divided into those used for soil physicochemical properties and those for microbial community analysis. Soil samples for the soil physicochemical properties’ analysis were air-dried in a cool place and debris was removed using a 2 mm mesh sieve; soil microbial community analysis was conducted in separate 50 mL conical tubes before these were dried and frozen at −20 °C before analysis.

### 4.2. Soil Physicochemical Property Analysis

Soil samples were air-dried and then sieved through a 2 mm mesh; gravel content was determined by weighing particles retained on the sieve, while the portion passing through the sieve was used for soil physicochemical property analysis. Soil texture analysis, which can measure the content of sand, mass, and clay in soil, was measured based on Stokes’s law [63]. The soil was analyzed according to the soil physicochemical analysis method proposed by the Rural Development Administration. Organic matter (OM) content was determined by the Walkley–Black method [64], and available phosphate (Av.P_2_O_5_) by the Lancaster leaching method with 1-amino-2-naphtol-4-sulfanic acid. The cation exchange capacity (CEC) was measured by the Kjeldahl distillation method, where NH4+ is substituted in the soil after leaching with 1-N ammonium acetate solution [65]. The cation content of Ca, K, Mg, and Na were measured using Inductively Coupled Plasma Optical Emission Spectrometry (ICP-OES) [66], and the acidity (pH) and electrical conductivity (EC) of the soil were measured using a pH meter and an EC meter, respectively, after diluting dry soil and distilled water 1:5.

### 4.3. Growth Characteristics of S. thea

To investigate the growth characteristics of *S. thea*, seven growth characteristics of leaves (length, width) and fruits (length, width, weight, sweetness, and hardness) were measured for each study site. The length and width of leaves and fruits were measured to the nearest 0.01 mm using a digital caliper (CD-200 APX, Mitutoyo Cp., Kanagawa, Japan), and the weight of fruits was measured to the nearest 0.01 g using a high-precision analytical balance (PAG214C, Ohaus Co., NJ, USA). The sweetness of the fruits was measured using a digital saccharimeter (PR-101α, Atago Co. LTD., Tokyo, Japan) to determine the soluble solid content of the fruit juice, and the hardness was measured using a physical property tester (CR-3000EX-S, Sun Scientific CO., Tokyo, Japan) with a probe diameter of 2 mm and a depth of 4 mm.

### 4.4. Soil DNA Extraction and PCR Amplification

The total DNA of each rhizosphere soil sample was extracted using the DNeasy Power Soil kit (QIAGEN, Hilden, Germany) following manufacturer instructions. After extraction, the quantification and quality of DNA were measured by PicoGreen and Nanodrop (Thermo Scientific, Rockford, IL, USA). Each sequenced sample was prepared according to the Illumina 16S Metagenomics Sequencing Library protocols (Macrogen, Seoul, Republic of Korea). In amplicon PCR, the V3-V4 region of the 16S rRNA gene of bacteria was targeted using the 16S V3-V4 primers [67]. The 16S V3-V4 primer sequences are as follows: 16S amplicon PCR forward primer, 5’-TCGTCGGCAGCGTCAGATGTGTATAAGAGACAGCCTACGGGNGGCWGCAG-3’; 16S amplicon PCR reverse primer, 5’-GTCTCGTGGGCTCGGAGATGTGTATAAGA GACAGGACTACHVGGGTATCTAATCC-3’. Input gDNA was amplified with 16S V3-V4 primers, and a subsequent limited-cycle amplification step was performed to add multiplexing indices and Illumina sequencing adapters. The conditions for amplicon PCR were as follows. First PCR: initial denaturation at 95 °C for 3 min, followed by 25 cycles of denaturation at 95 °C for 30 s, annealing at 55 °C for 30 s and extension at 72 °C for 30 s, and a final extension at 72 °C for 5 min. The condition for the index PCR was as follows. Second PCR: initial denaturation at 95 °C for 3 min, followed by 8 cycles of denaturation at 95 °C for 30 s, annealing at 55 °C for 30 s, and extension at 72 °C for 30 s, and a final extension at 72 °C for 5 min. The final products were normalized and pooled using PicoGreen, and the size of the libraries were verified using the TapeStation DNA screentape D1000 (Agilent, Santa Clara, CA, USA).

### 4.5. Pyrosequencing and Data Processing

Bacterial DNA sequencing was performed using the Illumina MiSeq™ sequencing system (Illumina Inc., San Diego, CA, USA) according to the manufacturer’s instructions. Raw sequences of bacterial DNA were processed using Mothur pipeline (version 1.43.0, The University of Michigan, Ann Arbor, MI, USA) [68]. The forward and reverse reads obtained from the Illumina platform were assembled, and sequences with a quality score <20 and ambiguous nucleotides were discarded before performing downstream analysis. The resulting sequences spanning the V3-V4 region were checked for the presence of chimeras using the function chimera.uchime. Taxonomic classification was performed using the “Greengenes reference database” for bacteria. Greengenes was used as it was reported to provide the best combination of speed and quality [69]. The sequences were clustered into operational taxonomic units (OTUs) at a 97% similarity level using a distance-based greedy clustering method (DGC) in Mothur. OTUs with less than 10 sequences were discarded to reduce false diversity.

### 4.6. Data Analysis

Data analysis was performed using SPSS software (Statistical Package for Social Sciences, Version 26, IBM SPSS statistics, Chicago, IL, USA), and data were expressed as means. The data of the results underwent one-way analysis of variance (ANOVA) and Duncan’s test, with statistical significance set at *p* < 0.05. And correlation coefficient analysis between soil physicochemical properties, growth characteristics, and soil microbial communities were analyzed using Pearson’s correlation.

## 5. Conclusions

This study analyzed the correlation between soil geochemistry, growth characteristics, and soil microbial communities of *S. thea* habitats. The study revealed significant correlations between soil chemistry and *Acidobacteria* and *Actinobacteria* phyla, and between *S. thea* growth characteristics and the *Chloroflexi* phylum. These results can provide a wealth of information about the optimal growing conditions for *S. thea*. And it is believed that with continued research, we will be able to establish more definitive cultivation techniques.

## Figures and Tables

**Figure 1 plants-13-03310-f001:**
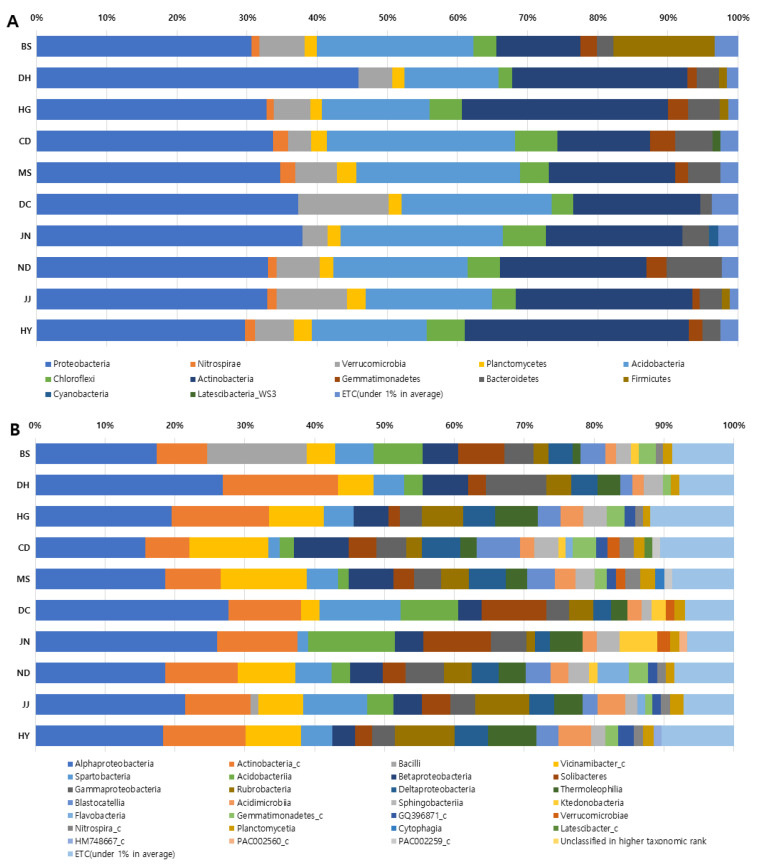
Clustering and relative abundance of microbial community in 10 different *S. thea* habitats. (**A**). Phylum; (**B**) classes; BS., Baeksan-ri, Sinan-gun; DH., Dohang-ri, Jindo-gun; HG., Hakga-ri, Haenam-gun; CD., Chungdo-ri, Wando-gun; MS., Mangseok-ri, Wando-gun; DC., Docheong-ri, Wando-gun; JN., Jangnam-ri, Goheung-gun; ND., Nangdo-ri, Yeosu-si; JJ., Jeoji-ri, Jeju-si; HY., Haye-dong, Seogwipo-si.

**Figure 2 plants-13-03310-f002:**
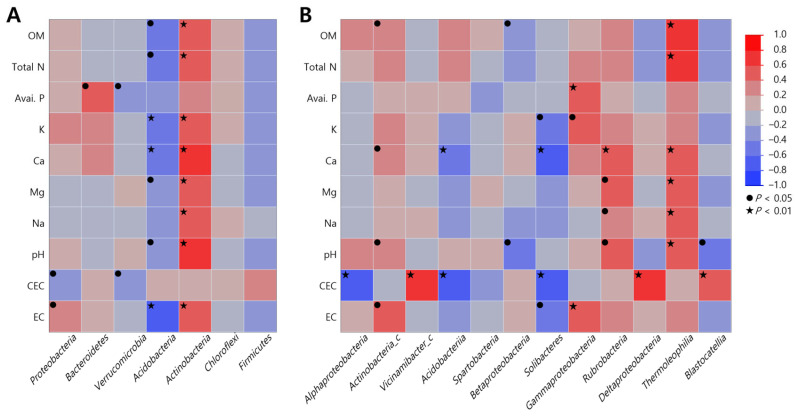
Duncan’s correlation analysis between soil physicochemical properties and microbial communities of *S. thea* habitats. (**A**) Phylum; (**B**) classes. Values in brackets indicate *p* value (● *p* < 0.05, ★ *p* < 0.01).

**Figure 3 plants-13-03310-f003:**
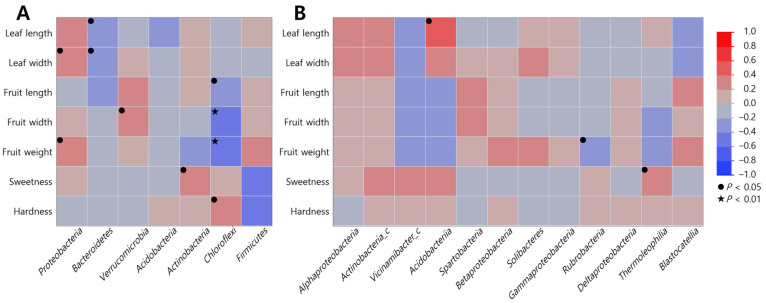
Duncan’s correlation analysis between growth characteristics of *S. thea* and soil microbial communities. (**A**) Phylum; (**B**) classes. Correlation coefficients (r) indicate significant correlations between variables compared. Negative values denote negative correlation and positive values denote positive correlation. Values in brackets indicate *p* value (● *p* < 0.05, ★ *p* < 0.01).

**Figure 4 plants-13-03310-f004:**
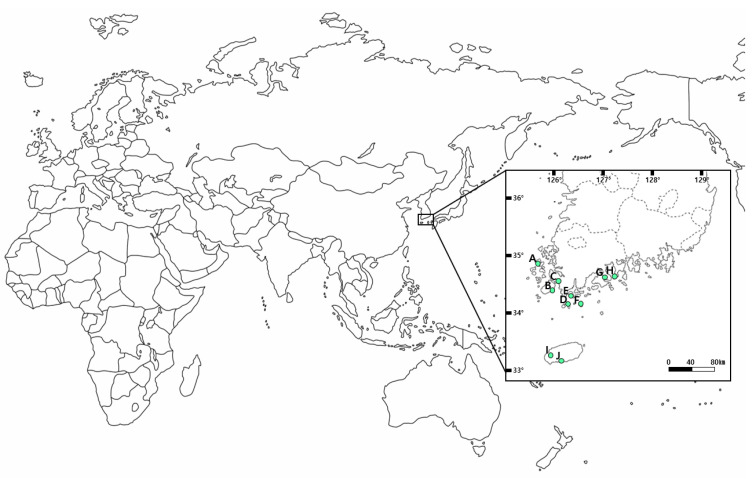
Map of collected sites of *S. thea* in Korea. A. BS (Baeksan-ri, Sinan-gun); B. DH (Dohang-ri, Jindo-gun); C. HG (Hakga-ri, Haenam-gun); D. CD (Chungdo-ri, Wando-gun); E. MS (Mangseok-ri, Wando-gun); F. DC (Docheong-ri, Wando-gun); G. JN (Jangnam-ri, Goheung-gun); H. ND (Nangdo-ri, Yeosu-si); I. JJ (Jeoji-ri, Jeju-si); J. HY (Haye-dong, Seogwipo-si).

**Table 1 plants-13-03310-t001:** Soil physicochemical properties of 10 *S. thea* habitats.

	BS	DH	HG	CD	MS	DC	JN	ND	JJ	HY
Sand	92.04	79.00	74.81	91.25	88.83	33.29	76.39	33.63	70.61	70.07
Silt	4.47	13.90	15.98	4.01	3.92	37.42	14.05	38.30	20.04	19.12
Clay	3.49	7.10	9.22	4.74	7.25	29.29	9.56	28.07	9.35	10.82
OM (%)	0.37	23.51	18.13	4.39	7.98	18.34	32.28	12.23	35.91	36.43
Total N (%)	0.03	0.90	0.60	0.23	0.27	0.57	1.21	0.48	1.30	1.39
Avai. P (mg/kg)	57.96	232.81	122.45	118.29	102.53	112.81	235.39	315.90	116.72	272.41
K (cmol^+^kg^−1^)	0.11	1.13	0.67	0.62	0.82	0.51	0.61	0.81	0.72	1.17
Ca (cmol^+^kg^−1^)	1.23	23.33	25.97	11.64	16.12	5.28	7.13	16.60	20.43	21.64
Mg (cmol^+^kg^−1^)	0.43	6.16	3.19	2.52	2.11	2.61	2.32	2.11	6.45	15.01
Na (cmol^+^kg^−1^)	0.04	0.37	0.28	0.14	0.13	0.33	0.16	0.22	0.51	6.16
CEC ^z^ (cmol^+^kg^−1^)	4.15	34.42	33.56	16.15	26.01	32.23	39.26	25.44	50.88	41.86
EC ^y^ (dS m^−1^)	0.11	1.78	0.84	0.57	0.59	0.54	0.64	0.68	1.16	1.59
pH [1:5, H_2_O]	6.21	5.76	6.42	7.05	6.45	5.37	4.68	6.14	5.63	6.75

BS., Baeksan-ri, Sinan-gun; DH., Dohang-ri, Jindo-gun; HG., Hakga-ri, Haenam-gun; CD., Chungdo-ri, Wando-gun; MS., Mangseok-ri, Wando-gun; DC., Docheong-ri, Wando-gun; JN., Jangnam-ri, Goheung-gun; ND., Nangdo-ri, Yeosu-si; JJ., Jeoji-ri, Jeju-si; HY., Haye-dong, Seogwipo-si. ^z^ cation exchange capacity, ^y^ electrical conductivity.

**Table 2 plants-13-03310-t002:** Growth characteristics of *S. thea* among 10 habitats.

Habitats	Leaf		Fruit				
	Length (mm)	Width (mm)	Length (mm)	Width (mm)	Weight (g)	Sweetness (°Brix)	Hardness (N)
BS	37.2 ^az^	21.49 ^ab^	7.24 ^ab^	8.45 ^a^	0.41 ^a^	11.1 ^d^	0.72 ^b^
DH	35.35 ^a^	21.08 ^ab^	6.89 ^bc^	8.4 ^b^	0.4 ^a^	14.63 ^bc^	0.67 ^b^
HG	34.82 ^ab^	22.55 ^a^	7.8 ^a^	8.83 ^a^	0.35 ^abc^	15.65 ^bc^	0.67 ^b^
CD	26.57 ^c^	16.98 ^bc^	7.15 ^ab^	7.94 ^abc^	0.34 ^abc^	12.19 ^cd^	0.94 ^b^
MS	35.38 ^ab^	20.44 ^ab^	7.2 ^ab^	8.17 ^ab^	0.39 ^ab^	18.91 ^a^	0.75 ^b^
DC	34.8 ^ab^	21.53 ^ab^	7.87 ^a^	8.95 ^a^	0.41 ^a^	15.76 ^abc^	0.82 ^b^
JN	38.15 ^a^	22.33 ^ab^	6.11 ^c^	6.77 ^d^	0.27 ^c^	17.79 ^ab^	0.84 ^b^
ND	27.54 ^bc^	15.02 ^c^	6.4 ^bc^	7.02 ^cd^	0.28 ^c^	15.35 ^abc^	1.56 ^a^
JJ	34.83 ^ab^	19.87 ^abc^	6.99 ^ab^	8.07 ^ab^	0.29 ^c^	14.87 ^bc^	0.84 ^b^
HY	36.72 ^a^	20.2 ^ab^	6.68 ^bc^	7.37 ^bd^	0.3 ^bc^	15.44 ^abc^	1.06 ^ab^

BS., Baeksan-ri, Sinan-gun; DH., Dohang-ri, Jindo-gun; HG., Hakga-ri, Haenam-gun; CD., Chungdo-ri, Wando-gun; MS., Mangseok-ri, Wando-gun; DC., Docheong-ri, Wando-gun; JN., Jangnam-ri, Goheung-gun; ND., Nangdo-ri, Yeosu-si; JJ., Jeoji-ri, Jeju-si; HY., Haye-dong, Seogwipo-si. ^z^ values followed by different letters within a column indicate significant difference (*p* < 0.05) between substrates for that parameter using DMRT (Duncan’s Multiple Range Test), (*n* ≥ 20, mean).

**Table 3 plants-13-03310-t003:** Collection of *S. thea* accessions from different locations in Korea.

Habitats	Administrative Distract			Coordinate	
	State	City	Town	North Latitude	East Longitude
BS	Jeollanam-do	Sinan-gun	Backsan-ri	34°52′44.8″	126°01′38.3″
DH	“	Jindo-gun	Dohang-ri	34°24′46.8″	126°18′17.2″
HG	“	Haenam-gun	Haksa-ri	34°24′21.7″	126°29′24.7″
CD	“	Wando-gun	Chungdo-ri	34°12′48.2″	127°36′16.9″
MS	“	“	Mangseok-ri	34°18′16.7″	127°44′59.0″
DC	“	“	Dicheong-ri	34°11′22.3″	127°34′32.9″
JN	“	Goheung-gun	Jangnam-ri	34°36′09.5″	127°24′56.4″
ND	“	Yeosu-si	Nangdo-ri	34°36′47.8″	128°32′16.8″
JJ	Jeju-do	Jeju-si	Jeoji-ri	33°19′31.9″	126°17′06.3″
HY	“	Seogwipo-si	Haye-dong	33°14′04.5″	126°23′53.1″

## Data Availability

The data presented in this study are available under permission from the corresponding author on reasonable request.

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
