# Peer review of "Correlation Analysis of Soil Microbial Communities and Physicochemical Properties with Growth Characteristics of Sageretia thea Across Different Habitats"

_plants, 2024, doi:10.3390/plants13233310_

Round 1
Reviewer 1 Report
Comments and Suggestions for Authors
As a reviewer for the paper entitled "Correlation Analysis of Soil Microbial Communities and Physicochemical Properties with Growth Characteristics of Sageretia thea across Different Habitats," I recommend that the paper requires some modifications. While the study presents significant and valuable findings, there are areas that could benefit from further refinement. I suggest the authors focus on improving certain aspects of the methodology, clarifying the data interpretation, and enhancing the overall clarity of the writing. With these adjustments, the paper has the potential to make a meaningful contribution to the field. My general and specific comments are given below;
Abstract
L13-14, the unit cmol+/kg please check this.
L21, Furthermore, is italic
L26-27, Remove all those keywords which is already present in title of the article
Introduction:
Please add specific literature related to the given plant family. The introduction is very general. The objective statement needs more clarification.
L71, woll….a typo error
Results:
L74-82, please check the units for different measurements
Table 1, check the units
Also perform PCoA for the data
Discussion:
Add a paragraph and discuss about the different metals interaction with microbial community and link your findings.
M & M:
Need to present detail methodology for soil physicochemical analysis

Need to improve the language and remove typo errors
Author Response
I've attached my review.
Thank you for your review.

Reviewer 2 Report
Comments and Suggestions for Authors
The aim of the research was to first mention the analysis of correlative relations, only later the investigate the growth characteristics of Sageretia thea (In the Abstract)
The abstract does not mention that the research was done in different areas, so it is not clear what the abbreviations JN, HG, DC mean.
Materials and Methods must be after Introduction.
Line 136 says that the correlation is negative, while the given coefficient r is positive.
The numbering of the figures does not meet the requirements.
According to the data in the article, I could not claim that plant growth depepended on microoganisms. The plants were affected by many different variables.
Author Response

(The authors gave the same response as above.)

Reviewer 3 Report
Comments and Suggestions for Authors
In general, the wording choices, tense alignment, and grammar should be addressed throughout the manuscript for clarity and readability. Also, specific changes would improve the manuscript including:
Rewording of the introductory paragraphs would be useful to more smoothly describe the state of the field and to engage the reader with the importance and purpose of the work.
The abstract could be modified to mention the major methodology utilized in the experiments.
Microorganisms is not hyphenated and the Discussion subtitle is misspelled.
Rewording of the Results section to include more references to specific methods as described in the Materials and Methods section will provide continuity and make the Results section a more organized reading experience.
Beginning in Section 2.2, the units designation “mm” appears to be formatted differently than the other text around it—not sure what is causing that.
Perhaps add language to clarify the physiochemical analysis of soil method of Rural Development Administration (at line 301). Is this a guidance document that proposes the group of methods described in that paragraph for assessing organic matter, cation exchange capacity, metals content, pH, and EC?
Comments on the Quality of English LanguageRewording is needed throughout the manuscript to avoid problems with language usage distracting from the work. This is especially important in the Introduction and Results section.
Author Response

(The authors gave the same response as above.)
